# Species-Specific Impact of *Fusarium* Infection on the Root and Shoot Characteristics of Asparagus

**DOI:** 10.3390/pathogens9060509

**Published:** 2020-06-24

**Authors:** Roxana Djalali Farahani-Kofoet, Katja Witzel, Jan Graefe, Rita Grosch, Rita Zrenner

**Affiliations:** Plant-Microbe Systems, Leibniz Institute of Vegetable and Ornamental Crops (IGZ) e.V., 14979 Großbeeren, Germany; kofoetr@igzev.de (R.D.F.-K.); Witzel@igzev.de (K.W.); graefe@igzev.de (J.G.); grosch@igzev.de (R.G.)

**Keywords:** *Fusarium oxysporum* f.sp. *asparagi*, *Fusarium proliferatum*, *Fusarium redolens*, MALDI-TOF MS, morphology imaging, plant pathogen interaction, RT-qPCR, soil-borne fungal pathogens

## Abstract

Soil-borne pathogens can have considerable detrimental effects on asparagus (*Asparagus officinalis*) growth and production, notably caused by the *Fusarium* species *F. oxysporum* f.sp. *asparagi*, *F. proliferatum* and *F. redolens*. In this study, their species-specific impact regarding disease severity and root morphological traits was analysed. Additionally, various isolates were characterised based on in vitro physiological activities and on protein extracts using matrix-assisted laser desorption ionisation time-of-flight mass spectrometry (MALDI-TOF MS). The response of two asparagus cultivars to the different *Fusarium* species was evaluated by inoculating experiments. Differences in aggressiveness were observed between *Fusarium* species and their isolates on roots, while no clear disease symptoms became visible in ferns eight weeks after inoculation. *F. redolens* isolates Fred1 and Fred2 were the most aggressive strains followed by the moderate aggressive *F. proliferatum* and the less and almost non-aggressive *F. oxysporum* isolates, based on the severity of disease symptoms. Fungal DNA in stem bases and a significant induction of pathogenesis-related gene expression was detectable in both asparagus cultivars. A significant negative impact of the pathogens on the root characteristics total root length, volume, and surface area was detected for each isolate tested, with Fred1 causing the strongest effects. No significant differences between the tested asparagus cultivars were observed.

## 1. Introduction

*Asparagus officinalis* L. is a perennial horticultural crop grown over a wide range of soils and under various climatic conditions. It can remain profitable for about 10 years in temperate regions that have long growing seasons. However, a gradual decline in productivity and the growth of asparagus plants can be observed within the growing period due to the ‘asparagus decline’ syndrome [1] mainly associated with the elevated soil densities of pathogenic *Fusarium* spp. [2]. Additionally, ‘early decline’ and ‘asparagus replant disease’ can occur when previous asparagus fields are replanted. Fusarium crown and root rot disease affects almost all plant organs. *Fusarium oxysporum* (Schltdl.) f.sp. *asparagi* and *Fusarium proliferatum* (Matsush.) are considered the main pathogenic species causing asparagus rots worldwide [3,4,5,6,7] and therefore have been studied the most. Depending on the continent and country, there are further *Fusarium* species involved, such as *Fusarium redolens* (Wollenw.) and *Fusarium solani* (Mart.) in Europe [8]. Here, *F. redolens* is of interest because this species is closely similar to *F. oxysporum* [9]. *F. oxysporum* enters its host through the root and induces either root rots or tracheomycosis when invading the vascular system [10]. Symptoms on asparagus caused by different *Fusarium* species are similar and include brown lesions on roots, and stems as well as rots of roots, rhizomes and stem bases. These damages lead to the stunting of plants, reduced plant size, chlorosis of fern stalks and wilt, and finally, resulting over time in weakening of crowns or loss of crowns [1,3,4,11].

Control of Fusarium disease in commercial fields of asparagus is very complex because of the deep and wide spread root system, the ubiquitous presence of *Fusarium* in the environment, and its systemic propagation in the host. In addition, the persistence of the *Fusarium* spp. in plant debris and in soil over many years, due to the formation of resting organs, hampers efforts to control the disease. Thus, fungicidal application is inefficient and affects the environment negatively. Sustained management strategies to minimise Fusarium crown and root rot in asparagus cultivation comprises, on one hand, inoculum reduction and soil environment alterations and, on the other hand, the enhancement of the host defence potentials by developing robust plants [8]. The latter strategy includes three main approaches: breeding of resistant cultivars, strengthening plant health by inducing resistance in plants via biocontrol methods, and stimulation of root health by colonizing asparagus roots with arbuscular mycorrhizae [8]. Approaches for the breeding of *Fusarium*-resistant cultivars are difficult because of the perennial nature of asparagus, its lack of genetic uniformity (dioecious plants), and the broad genetic diversity of *Fusarium* in its interaction with the host [2,12,13,14,15,16,17]. There have been some efforts to manage Fusarium crown and root rot of asparagus by the use of non-pathogenic *Fusarium* strains or other biological control agents, such as *Trichoderma harzianum,* to induce systemic acquired resistance, but so far the success in these efforts was moderate [2,18]. The stimulation of root health by colonizing asparagus roots with arbuscular mycorrhizae has been more successful against *Fusarium* attack [2,19,20,21,22]. 

Detailed information about the response of asparagus during the interaction with *Fusarium* spp. at the molecular and metabolic level are lacking at present. However, such knowledge can support future breeding programmes or the development of plant protection strategies. The occurrence of *Fusarium* spp. is currently identified using molecular markers and phylogenetic analysis mainly based on sequences of the transcription elongation factor gene *TEF-1a* [23,24,25]. In asparagus *Fusarium* species-specific primers for PCR analysis have been developed [26,27], and the genetic diversity analyses by PCR-denaturing gradient gel electrophoresis [28], and single-stranded conformational polymorphism [29]. Corpas-Hervias et al. demonstrated, via random amplified polymorphic DNA-polymerase chain reaction (RAPD-PCR), the affiliation of *F. proliferatum* and *F. moniliforme*, renamed to *F. verticillioides*, to one group [30]. Amplified fragment-length polymorphism fingerprinting identified *F. redolens* as a pathogen involved in root, crown and spear rot of asparagus [9]. Nowadays, occurrence of *F. redolens* is also tested using molecular markers [31]. The presence of different *Fusarium* species isolated from fields with Fusarium crown and root rot history showed considerable variability among the isolates using intersimple-sequence repeat PCR analysis [7]. 

Root growth is an important indicator for the development and progress of diseases induced by soil-borne pathogens [32,33]. The attack or frequency of host infection by pathogens is influenced by root growth traits such as root length, diameter, surface area, and volume. Tomato plants inoculated with *F. oxysporum* f.sp. *lycopersici* showed decreased root lengths and reduced root weight, root surface area and root volume [34]. The perennial asparagus builds an extensive root system that comprises thick, long, storage roots clothed with fine, fibrous, absorptive roots that can be exposed to pathogenic soil-borne *Fusarium* strains [3]. It has already been reported that *F. oxysporum* f.sp *asparagi* causes severe foot and root rot, thus significantly reducing root weight [11]. In this regard it is essential to investigate the characteristics of the root system of asparagus after being attacked by *Fusarium* species at the early stages of plant growth and plant pathogen interaction.

So far, the impact of different *Fusarium* species on the root system and root characteristics of asparagus remains unclear. In this work, we evaluated the response of *asparagus* to the *Fusarium* species *F. oxysporum* and *F. proliferatum*, which are known to be ubiquitous in the soil of asparagus-growing areas affected by Fusarium crown and root rot [3], and to *F. redolens*, as the species is closely similar to *F. oxysporum* [9]. The selection of two asparagus cultivars was based on their differences in growth and robustness, according to the breeders’ assessments, as differences in response to *Fusarium* infection were assumed. Thus, the objectives of this study were to (i) characterise isolates of *F. oxysporum*, *F. proliferatum* and *F. redolens* at physiological, pathogenic and molecular levels; (ii) identify the defence response of two asparagus cultivars to isolates of the three *Fusarium* species by verifying fungal DNA presence in infected plants and analysing the expression of defence-related genes by RT-qPCR; and (iii) study root and shoot characteristics and identify root morphology characteristics of two asparagus cultivars in response to infection by the three *Fusarium* species using an automated analysis of root images. 

## 2. Results

### 2.1. Identification and Physiological Characterisation of Fusarium Spp.

In order to analyse whether the efficacy in degrading plant-derived carbon sources is correlated with disease severity caused by the *Fusarium* spp. isolates, we tested their ability to utilise different plant-derived polysaccharides. Fungal growth on complex polysaccharides and cell wall constituents, cellulose and pectin was slower as compared to the other tested substances (Figure 1). No species-specific growth response was found for cellulose as carbon source. However, *F. oxysporum* isolates showed better growth on pectin as compared to the other *Fusarium* species. In addition, extracellular enzymatic activities were analysed in plate tests as another factor influencing the usability of these polysaccharides as carbon source for fungal growth. By visualisation on chromogenic medium, no extracellular enzyme activity was detected when *Fusarium* spp. were cultivated on the two substrates cellulose and pectin. In contrast, extracellular enzymatic activity was detected on plates containing starch, trehalose, and cellobiose in all tested isolates, with the latter substance being degraded most efficiently (Appendix A). Substrate utilisation of trehalose and starch was significantly higher in *F. oxysporum* isolates as compared to *F. proliferatum* and *F. redolens* (Figure 1).

To test the potential of matrix-assisted laser desorption ionisation time-of-flight mass spectrometry (MALDI-TOF MS) for the species characterisation of *Fusarium* isolates, mass spectra on the basis of protein fingerprints of mycelium from overnight fungal cultures were obtained. A comparison of spectra from the different *Fusarium* spp. by hierarchical clustering showed a clear separation between *F. proliferatum* isolates and the remaining isolates, while no separation between *F. oxysporum* and *F. redolens* isolates was found (Figure 2).

### 2.2. Aggressiveness of Fusarium spp. Isolates on Asparagus Plants

Possible differences in the aggressiveness of *Fusarium* isolates and the susceptibility of the two asparagus cultivars, ‘Backlim’ and ‘Rapsody’, were initially tested by monitoring disease symptoms and plant growth. The typical symptoms on roots were brown to reddish-brown necrotic spots and lesions of variable length on both storage and fibrous roots. The tissue within the rhizome was strongly affected so that the inner tissue of storage roots was degraded and roots consisted only of root rhizodermis and the central cylinder. This was predominantly observed by inoculation with *F. proliferatum* and *F. redolens* isolates (Appendix A). Control plants did not show any symptoms. Two-way ANOVA of DS of roots showed no interaction between cultivars and isolates. Therefore, the influence of isolates on DS is statistically analysed across both cultivars. However, with regard to the biological difference of ‘Backlim’ and ‘Rapsody’ in plant performance, we show the effect of isolates on the DS of roots separately for each cultivar, being aware that statistical significance between isolates and control counts for both cultivars likewise. This type of presentation allows the comparison between treatments differing both in cultivar and isolate level [35]. Disease severity (DS) ratings of roots inoculated with *Fusarium* spp. isolates differed significantly from the control plants (Figure 3 and Appendix A). DS ratings showed a significant species-specific aggressiveness on the roots of both cultivars: *F. redolens* was the most aggressive species followed by *F. proliferatum* and thereafter *F. oxysporum*. Between the cultivars ‘Backlim’ and ‘Rapsody’ no significant difference in susceptibility of roots was observed. DS values were significantly discriminable between inoculation with the isolates of the three *Fusarium* species on ‘Backlim’ and ‘Rapsody’; except for isolate Fpro3 that had similar impact like Foa1 and Foa4. Among the *F. oxysporum* f.sp. *asparagi* isolates Foa1 and Foa4 were likewise aggressive and more aggressive than Foa2 and Foa3 but they still did not cause DS scores higher than 2 and fell below the normalised rank mean of 0.5 (Figure 3). No difference in aggressiveness was observed among the *F. proliferatum* isolates Fpro1 and Fpro2, with both causing higher DS than Fpro3 (Figure 3). The highest DS ratings on both cultivars were caused by the two *F. redolens* isolates Fred1 and Fred2, which differed significantly from each other. In contrast to root, the DS ratings of ferns were very low, with highest value of 1.9, and almost no differences among the *Fusarium* spp. (Appendix A). The DS of ferns did not differ significantly between the *Fusarium* species, and the correlation between the DS of roots and of stems was not significant (r^2^ = 0.347). 

Two-way ANOVA of root fresh weight (FW) revealed no interaction between cultivars and isolates. We found a significant main effect of cultivar and isolate on root FW accumulation; root FW between the cultivars was significantly different. Thus, mean values of cultivars are depicted separately and statistical significance between isolates and control counts for both cultivars likewise. Root FW after inoculation with all isolates was significantly lower than the control. Among all isolates, Foa2, Foa3, and Foa4 had significantly lowest impact on FW reduction compared to Fpro3, Fred1, and Fred2. The highest impact was observed on plants inoculated with *F. redolens* isolate Fred1 (Figure 4A).

Two-way ANOVA of fern FW revealed no interaction between cultivars and isolates. We found significant main effect of cultivar and isolate on fern FW accumulation; fern FW between the cultivars differed significantly (Figure 4B). Thus, mean values of cultivars are depicted separately and statistical significance between isolates and control counts for both cultivars likewise. Compared to the control, a significant reduction in fern FW was only observed when plants were inoculated with *F. proliferatum* isolate Fpro3 and *F. redolens* isolates Fred1 and Fred2 (Figure 4B). A medium but significant correlation was observed between the DS of roots and root FW (r^2^ = 0.539), and between root DS and fern FW (r^2^ = 0.533). The correlation between the FW of root and fern was highly significant (r^2^ = 0.835).

### 2.3. Molecular Detection of Fusarium spp. in Stem Tissue

The specific detection of *Fusarium* spp. was carried out in the stem bases of harvested plants to provide evidence of fungal spread in inoculated plants. Species-specific primers were based on sequence variations, previously analysed and standard curves were calculated in the presence of plant DNA as described. Although proof of presence of the specific fungal DNA was clear evidence of infection, the amount of fungal DNA did not necessarily correlate with the severity of disease symptoms. However, a certain relationship between fungal DNA in the stem bases of asparagus and root FW after inoculation with *Fusarium* spp. was detectable in both cultivars ‘Backlim’ (r^2^ = 0.402) and ‘Rapsody’ (r^2^ = 0.690) (Appendix A). 

### 2.4. Expression of Defence-Related Genes

Quantitative real-time reverse transcription PCR was performed to analyse the RNA accumulation of defence genes coding for pathogenesis-related proteins. Differences in the relative expression levels of pathogenesis-related protein 1 (*PR1-2*), pathogenesis-related protein 1-like (*PR1-4*), cationic peroxidase 1-like (*POX*), and phenylalanine ammonia-lyase (*PAL*) in the stem bases of harvested plants are shown in Figure 5. While the steady-state levels of mRNA coding for *PAL* were unchanged in both cultivars 8 weeks after inoculating roots with *Fusarium* spp. isolates, a significant reduction in the steady-state levels of mRNA of *POX* was detectable in ‘Rapsody’ after inoculation with *F. oxysporum* and *F. redolens* isolates. An induction of *PR*-genes was found in both cultivars, with the strongest increase in relative gene expression after inoculation with *F. redolens.* The expression increase of *PR1-2* was significant and higher in ‘Backlim’ than in ‘Rapsody’.

### 2.5. Changes in Root Morphology

Based on the significant disease symptoms in the roots of asparagus plants that were inoculated with *Fusarium* spp. isolates, a more quantitative investigation of root morphology was performed using image analysis. Two-way ANOVA revealed no interaction between cultivars and isolates. Statistically significant difference between isolates and control counts for ‘Backlim’ and ‘Rapsody’ likewise. Because of biological cultivar specification, also, here, the mean values of cultivars are depicted separately. No difference between cultivars was observed by none of the traits except for fibrous root radius. The inoculation of asparagus plants with the nine *Fusarium* spp. isolates was associated with a decrease in total length of both, fibrous and storage roots, as measured separately (Figure 6). Total root lengths of plants inoculated with *F. oxysporum* isolate Foa1, *F. proliferatum* isolates Fpro1 and Fpro3, and *F. redolens* isolates Fred1 and Fred2 were significantly lower than control plants (Figure 6). Isolate Fred1, provoking the highest DS among the nine tested *Fusarium* spp. (Figure 3), caused a storage root length reduction of 46% compared to control plants. A significant reduction of storage root length of 36 and 40% was also observed when plants were inoculated with Fpro1 and Fpro3 (Figure 6).

*F. redolens* isolate Fred1 significantly reduced the fibrous root lengths of both cultivars and induced a decrease of approximately 77% (Figure 6). In addition, Fred2 caused a reduction of fibrous loot length by 63%. Among the *F. proliferatum* isolates Fpro1 and Fpro3 caused significant fibrous root length reductions of ca. 50 and 60%. *F. oxysporum* isolate Foa1 had a significant effect on plants when compared to controls. DS of roots correlated significantly with fibrous root length (r^2^ = 0.499), but not with the storage root length (r^2^ = 0.406). High correlation values between root length and fern FW (r^2^ = 0.759), and between root length and root FW (r^2^ = 0.890) show a close relation between these two traits. 

No impact of *F. oxysporum* was observed on the root radius, subdivided into fibrous and storage roots radius (Appendix A). *F. proliferatum* isolate Fpro3 caused a significant reduction in storage and fibrous roots’ radius

As a consequence of decreased root lengths, the fibrous and storage roots’ volumes of plants that were inoculated with the *Fusarium* spp. isolates Foa1, Fpro1, Fpro2, Fpro3, Fred1, and Fred2 were significantly reduced (Figure 7). 

The root surface area of ‘Backlim’ and ‘Rapsody’ control plants was 850 and 875 cm^2^, respectively (Figure 8). Inoculation with Fpro1, Fpro3, Fred1 and Fred2 reduced root surface area to nearly half of that of the control plants. The utmost reduction about more than 500 cm^2^ was caused by inoculation with *F. redolens* isolate Fred1. *F. oxysporum* f.sp. *asparagi* isolate Foa1 and Fpro2 also had a significant impact on the surface area reduction of plants (Figure 8). The correlation values between root surface area and root DS, root FW and stem FW (r^2^_root DS_ = 0.583; r^2^_root FW_ = 0.908; r^2^_stem FW_ = 0.689) indicate a tense relation between these traits.

## 3. Discussion

Asparagus plants can be infected by different *Fusarium* species and are colonised via the roots. The influence of *Fusarium* on the root growth characteristics of asparagus depending on the species has not been studied yet. In this study, the impact of four isolates of *F. oxysporum*, three of *F. proliferatum* and two of *F. redolens* was analysed on the asparagus cultivars ‘Backlim’ and ‘Rapsody’, with different growing parameters. Hence, we expected differences in response to *Fusarium* infection depending on the cultivars.

Since different *Fusarium* spp. can occur on asparagus, the availability of a rapid and safe species-specific diagnosis tool would be useful. Nowadays, the identification of microbial pathogens is routinely performed using matrix-assisted laser desorption ionisation time-of-flight mass spectrometry (MALDI-TOF MS, Bruker Daltonik, Bremen, Germany ), especially for bacterial iso lates [36]. Additionally, mass-spectrometry-based approaches have also been applied successfully for the identification and differentiation of *Fusarium* species based on spore protein profiles [37,38]. Furthermore, MALDI-TOF MS peptide/protein fingerprints have been used to distinguish closely related species in the *Fusarium fujikuroi* species complex [39], and for in situ diagnosis in infected plant tissue [40]. Here, we have tested the potential of MALDI-TOF MS to distinguish the used *Fusarium* species based on protein fingerprints of mycelium. A reliable separation between *F. proliferatum* and the remaining *F. oxysporum and F. redolens* isolates was found. A comparable pattern of similarities among these *Fusarium* spp. was shown by Baayan et al., who performed AFLP fingerprinting followed by hierarchical clustering [9]. With an increasing database of fungal protein fingerprints, the future application of this rapid and economic tool for pathogen diagnosis should lead to the establishment of better and, also, in situ fungal pathogen detection.

The comparative analysis of translated fungal genomes revealed that the highest number of carbohydrate-active enzymes occurs in necrotrophic and hemibiotrophic plant pathogenic fungi, with *Fusarium* spp. ranking among the highest [41]. The synthesis of cell wall degrading enzymes, a subset of carbohydrate-active enzymes, is of high importance during plant infection with *Fusarium* spp. [42], as some of those are secreted to facilitate infection and to exploit the plant carbon source. In this study, we intended to answer the question if differences in the activity of carbohydrate-active enzymes between the tested *Fusarium* spp. are given. It is to assume that a relation between the ability to degrade plant-derived carbon sources and aggressiveness exist. A species-specific substrate utilisation was found for pectin, trehalose and starch, although the response was contrarious to what was found for isolates of *F. culmorum* [43,44]. The least aggressive *F. oxysporum* isolates showed the highest ability to degrade pectin and to utilise starch and trehalose, indicating that other factors, despite the tested carbohydrate-active enzyme activities account for species-specific aggressiveness to asparagus. 

The impact of *F. oxysporum*, *F. proliferatum* and *F. redolens* isolates was investigated at an early growth stage of asparagus with particular emphasis on changes in the root morphology. Based on the present pathogenicity test, we found that our tested isolates differed significantly in their aggressiveness, and, to a certain extent, a classification of the *Fusarium* spp. in four pathogenicity groups was possible. *F. redolens* isolates were the most aggressive strains followed by the moderate aggressive *F. proliferatum* and the weakly aggressive *F. oxysporum* isolates. Based on the severity of disease symptoms, significant differences in aggressiveness were observed between the isolates of each species. Following this classification, the *F. proliferatum* isolate Fpro3 should rather be ranged in the group of the two pathogenic *F. oxysporum* isolates Foa1 and Foa4. Following the argumentation of Elmer and Stephens [45] who considered isolates causing DS higher than 20% as pathogenic on asparagus, the two isolates Foa2 and Foa3 that were not able to cause DS scores higher than 1 (that includes a diseased root area up to 20%) could be rated as non-pathogenic. However, it has to be noted that our testing was performed with rather young plants and long-term observations in fields may reveal stronger and increasing impact over time. It is worth mentioning that, in the past, many pathogenicity tests with *Fusarium* strains performed in different countries resulted in diverse outcomes. Borrego-Benjumea et al. [7] showed in their pathogenicity tests that *F. oxysporum*, *F. solani* and *F. acuminatum* were highly aggressive, whereas *F. proliferatum* and *F. redolens* were weakly aggressive. The most aggressive *Fusarium* spp. in Spain were *F. proliferatum* and *F. solani* [30], and, in the Netherlands, Blok and Bollen [11] found, in *F. oxysporum* isolates, the highest aggressiveness. In Japan and the United Kingdom, *F. oxysporum* and *F. proliferatum* were the most pathogenic *Fusarium* spp. responsible for asparagus decline [29,46]. In summary, this demonstrates that the assessment of pathogenicity depends on the analysed isolates, the cultivars, and the interaction of both with the environmental conditions.

The results in the present experiments show a comparable level of susceptibility in the two cultivars, while they were chosen because of differences in growth and robustness. There was no cultivar-specific significant difference in the DS of roots caused by the various *Fusarium* spp. Thus, both cultivars can likely be regarded as equally susceptible to at least the tested *Fusarium* spp. isolates. 

The aggressiveness of the isolates of the three *Fusarium* species is largely reflected in the root traits. Compared to control plants, the root FWs of plants were reduced when they were inoculated with each isolate. A decrease in root FW was noticed with increasing DS of roots. These results are in accordance with the findings of Corpas-Hervias et al. [30], who found a general trend in the decrease in root dry weight with increasing severity of root rot symptoms. An impact of the isolates on fern FW was obvious when plants were inoculated with *F. proliferatum* isolate Fpro3 and *F. redolens* isolates Fred1 and Fred2, with Fred1 being the most aggressive isolate. The impact of the *F. oxysporum* f.sp. *asparagi* isolates was low, highlighting the weak aggressiveness of this *Fusarium* species, but also Fpro1 and Fpro2 had only weak impact on the FW of fern. 

In our experiments, the DS of ferns was very low and we observed no significant impact of the fern DS on root FW (r^2^ = 0.198) or on fern FW (r^2^ = 0.347), and also no significant correlation existed between the DS of roots and DS of ferns (r^2^ = 0.347). This is in contrast to the results of Corpas-Hervias et al. [30], who found a highly significant correlation between root rot and fern severity ratings. It is assumed that the rather short duration of our experiment with plant harvest eight weeks post inoculation may have minimised the impact of the pathogens on fern FWs, although the DNA of the pathogen was already detectable in the stem bases. The presence of fungal DNA clearly provides proof of plant infection; however, the amount of fungal DNA measured in the respective tissue does not necessarily correlate with DS. Since we have inoculated plant roots, the detection of fungal DNA in stem bases is an expression of fungal spread into aerial parts of the plant. This is the typical strategy of soil-borne fungal pathogens that cause vascular wilt in host plants. After expansion in the host they produce resting structures in plant tissues that get incorporated into soil; from there they again start interaction by first challenging the roots and subsequently spread inside the vascular tissue [47]. In addition, the first evidence of the ferns’ response to the presence of the pathogen was detected using gene expression analysis. The expression of *PR1* was significantly higher and the expression of another pathogenesis-related gene (*PR9*) *POX* was unchanged or even reduced in stem bases, thus resembling a typical *PR*-gene expression profile upon *Fusarium* infection [48]. The expression of a key biosynthetic enzyme, mainly involved in defence mechanisms, that catalyses the first step in the synthesis of a variety of polyphenol compounds *PAL* [49] remained unchanged. This is in contrast to other experiments, where the presence of *F. oxysporum* f.sp. *lycopersici* strains induced *PAL* expression as a clear evidence of induced systemic acquired resistance in tomato [50]. Symptoms on ferns and the reduction of fern FW seems to be the result of a longer period of interaction between plant and the soil-borne *Fusarium* spp., and it would be interesting to analyse the impact of the pathogens on aboveground parts of plants at this later stage of pathogenesis.

The morphology of inoculated asparagus roots classified in storage and fibrous roots was measured by means of a new automated processing pipeline. No significant difference between ‘Rapsody’ and ‘Backlim’ was observed regarding these traits, except for fibrous roots radius. A general tendency in the decrease in root length could be observed with increasing severity of root symptoms. The measured root volume values showed a general tendency like the results of the total root length, and no clear conclusion could be drawn regarding data of radius of fibrous and storage roots. We assume that most probably after attack with the pathogen, particularly fibrous roots die off and hence the remaining roots that were measured were almost not injured and stayed unaffected by the pathogen. Additionally, symptoms caused by *Fusarium* spp. on affected roots comprised lesions that were not yet so severe on root tissue at assessment time and therefore root radius was less influenced. The decrease in total root length resulted in a reduction of root surface area. Apart from *F. proliferatum* isolates Fpro1, Fpro2, and Fpro3, and both *F. redolens* isolates Fred1, and Fred2 also *F. oxysporum* f.sp. *asparagi* isolate Foa1 caused a clear reduction in the root surface area. Altogether, it can be concluded that the two *F. redolens* isolates had the highest impact followed by *F. proliferatum* isolates Fpro1 and Fpro3. *F. oxysporum* f.sp. *asparagi* isolates had the lowest effect on root traits and fresh weight. Thus, the variation in aggressiveness of the isolates was to some degree reflected in the effect of the isolates on fresh weight and root morphology.

These observations on root morphology are in accordance with other studies with *Fusarium* spp. and further soil-borne pathogens. For example, a decrease in root length, root weight, root surface area, and root volume was observed on tomato plants inoculated with *F. oxysporum* f.sp. *lycopersici* [34]. Crosby [51] stated that total root length, fine and small root length, and the number of root tips were greater for a melon cultivar that was tolerant to the soil-borne fungus *Monosporascus cannonballus*. On the other hand, there are also opposed statements on the correlation between root traits and the impact of soil-borne pathogens. For example, Cichy et al. [52] could not find significant interactions between attacks with *F. solani* f.sp. *phaseoli* and the root system traits of beans. They made the short experiment duration of 32 days responsible for the missing impact of the pathogen on the host. In a soil compaction–*F. solani* root rot interaction study with large- and medium-rooted pea lines, Kraft and Boge [53] asserted that larger-rooted pea lines had significantly more roots than did the smaller-rooted line. However, they observed that reduction in root growth was stronger for the larger-rooted lines than for the smaller-rooted pea line.

Insights into the root morphology of plants are fundamental for the comprehension of root functionality. Fibrous roots of asparagus are mainly responsible for water and nutrient uptake and the storage roots have the role as storage locations for reserve carbohydrates (mostly fructans). The considerable decreases in fresh weight, root length, root volume, and root surface area as shown in this study entail the limitation of water and nutrient uptake of diseased plants and result in weakness and stunted growth of plants. The ability of asparagus plants to store carbohydrates in these roots is limited when storage roots are reduced and this has a negative effect on the production of spears for the growing season and hence on yield. In this study, we could clearly differentiate the aggressiveness of *Fusarium* spp. isolates. Moreover, the measured root traits, including root fresh weight, total root length, root volume, and root surface area, turned out to be meaningful indicators for the general status of the plant. The correlations between the severity of root rot symptoms and root traits were mostly significant. Arias et al. [54] observed for some *Fusarium* spp. isolates significant linear relationships between soybean yield and root symptoms and concluded from their studies that the root morphological characteristics of soybean were more consistent indicators of yield loss than root rot severity caused by *Fusarium* spp. on soybean. The yield loss in asparagus that is related with the asparagus decline is also related to the presence of *Fusarium* spp. in soil, but following Elmer [55] apart from this soil-borne pathogen there are many other indicators that contribute to asparagus decline. Guo [56] specified that asparagus cultivars with high yield potential exhibit a big root-stem-ratio and underlined the relevance of root mass for yield. To promote vigorous root systems, proper balance of the nutrient elements N, P, and K is essential particularly to help plants to withstand biotic and abiotic stress. By means of management practices to overcome poor nutrition, abiotic stressors for the implication of asparagus decline, such as water insufficiency and malnutrition, can be resolved [55]. In this short-time experiment, we could not prove the consequences of limited water and nutrient uptake, but we show that the root system is heavily restricted when roots are attacked by the pathogenic *Fusarium* spp. and poor nutrition can also be initiated by diseased root systems that are restricted in water and nutrient uptake. This is why the prevention of lesions and injuries of the root system of asparagus is a main concern. 

Most *Fusarium* species synthesise toxic secondary metabolites, known as mycotoxins, that are defined as evoking a toxic response when introduced in low concentrations to animals. It was shown for isolates of *F. proliferatum* from asparagus spears origin to produce moniliformin and fumonisin B1, and *F. oxysporum* to produce only moniliformin [57,58]. Moreover, the respective toxins can be found in basal parts of asparagus spears that were infected with the respective *Fusarium* species [59,60]. However, environmental factors affecting mycotoxin biosynthesis in the fungus and the roles of such mycotoxins in the infected plants are not yet fully understood [61,62]. The species-specific biosynthesis of such mycotoxins in the *Fusarium* asparagus interaction and its impact on root and shoot characteristics of asparagus plants should be the topic of a targeted future analysis.

## 4. Materials and Methods

### 4.1. Fusarium spp. Cultivation

The *Fusarium* spp. isolates used in this study are listed in Table 1. The isolates were provided either on synthetic nutrient-poor agar [52] or on potato-dextrose agar (PDA; Potato Dextrose Agar, Carl Roth GmbH, Karlsruhe, Germany) plates. Single-spore isolates were made of each isolate and cultivated on PDA in the dark at 25 °C for 10 days for the studies. 

### 4.2. Fungal Degradation of Polysaccharides and Production of Extracellular Hydrolytic Enzymes

The *Fusarium* spp. isolates were tested for their growth on plant-derived polysaccharides by growing them on cellobiose (Fluka Honeywell Specialty Chemicals, Seelze, Germany), trehalose (Carl Roth GmbH), Avicel (50 µm size cellulose, Sigma-Aldrich, Darmstadt, Germany), pectin (Sigma-Aldrich), or starch (neoLab, Heidelberg, Germany). The media contained 0.5% of respective polysaccharides, 0.1% yeast nitrogen base (Carl Roth GmbH), 0.05% congo red (Carl Roth GmbH) and 1.5% agar (VWR, Darmstadt, Germany). The plates (Ø 9 cm) were inoculated by placing, at the centre of the plate, an agar plug that was taken from the rim of 10-day-old actively growing colonies of each isolate on PDA (Table 1). Isolates were incubated at 25 °C for 4 (cellobiose, trehalose, and starch) or 7 days (cellulose, pectin) on the respective agar media. Seven days of incubation on cellulose and pectin was chosen because of the slower growth of these media. The mycelial diameter was measured as an indicator for carbon utilisation, as well as the diameter of congo red discoloration as an indicator of the production of extracellular enzymes. Eight replicate plates were tested per isolate and per carbon source. 

### 4.3. Species-Discrimination by MALDI-TOF Mass Spectrometry

Overnight cultures of *Fusarium* spp. isolates (Table 1), grown in potato-dextrose broth at room temperature under shaking conditions, were pelleted by centrifugation. Proteins contained in the mycelia were extracted using formic acid-based method [63]. A total of 1 µL of protein extract was spotted onto a polished steel target (Bruker Daltonik, Bremen, Germany) and air-dried before the addition of a saturated α-cyano-4-hydroxycinnamic acid solution as matrix. The MALDI method was calibrated using a bacterial test standard (Bruker Daltonik).

An ultrafleXtreme MALDI-TOF mass spectrometer (Bruker Daltonik), working in linear positive mode, was used for acquiring mass spectra in the range of *m/z* 200–20,000. Measurements were performed by flexControl v3.4 software (Bruker Daltonik). The MALDI Biotyper v3.1 software (Bruker Daltonik) was used to process the raw spectra, perform a database search against the Bruker Filamentous Fungi Library v1.0 (364 entries) for isolate identification, and to generate a hierarchical clustering of *m/z* values. Species-discrimination was performed three times with independently grown fungal cultures. 

### 4.4. Plant Cultivation

Seeds of *Asparagus officinalis* L. ‘Backlim’ (Limgroup B.V., AE Horst, the Netherlands) and ‘Rapsody’ (Südwestdeutsche Saatzucht GmbH & Co. KG, Rastatt, Germany) were surface sterilised in water at 55 °C for 20 min. Then, they were sown in trays filled with substrate consisting of a 1:2 mixture of sterilised sand and standardised plant growth substrate of a volcanic clay and peat soil mixture (Fruhstorfer Erde type P; Havita, Vechta, Germany), and cultivated in a growth chamber at 23 °C with a 12-h photoperiod until plants had reached a BBCH growth stage between 12 and 13, describing the evolvement of the second or third stem [64]. Plants were cultivated in a growth chamber at 23/18 °C and 75/85% relative humidity in a day/night cycle with a 16/8 h photoperiod (400 µmol m^−2^ s^−1^). Plants were watered as required and after cladode development they were fertilised once a week with macro- and micro-nutrients (Ca(NO_3_)_2_ (4.18 g L^−1^); KNO_3_ (1.03 g L^−1^); KH_2_PO_4_ (0.35 g L^−1^); K_2_SO_4_ (0.43 g L^−1^); Mg(NO_3_)_2_ (0.51 g L^−1^); MgSO_4_ (0.63 g L^−1^), Fe chelate (0.5 g L^−1^)).

### 4.5. Plant Inoculation and Pathogenicity Testing of Fusarium Spp. Isolates on Asparagus Plants

Asparagus plants (‘Backlim’ and ‘Rapsody’) were carefully removed from the substrate. Roots were rinsed with tap water and prepared for inoculation with spore suspensions. The spore suspension of each isolate (Table 1) was made by flooding single-spore isolate cultures on PDA plates with sterile distilled water and filtering it through four layers of sterile cheesecloth. The number of conidia was microscopically counted (Zeiss Axioskop 2, Carl Zeiss Microscopy GmbH, Jena, Germany) using a haemocytometer and adjusted to 6–8 × 10^6^ conidia mL^−1^ by dilution with sterile distilled water. Plant inoculation was performed by dipping roots for 30 min in 10 mL of the spore suspension of the respective *Fusarium* isolate and adding 9 mL of the spore solution around the stem after the plants were transplanted singly in pots (10 × 10 × 11 cm, 0.69 L) containing sterile substrate (sand:Fruhstorfer Erde type P 1:1 (vol/vol)). The control plants were inoculated in the same manner but with sterile distilled water. After inoculation, plants were grown in growth chambers as described above until disease assessment. The experiment was performed three times. Each experiment included three replications for each isolate and cultivar. One replicate comprised six plants of the respective cultivar. Pots were placed in a randomised complete block design. Eight weeks after inoculation, *Fusarium* symptoms were assessed on roots and stems. Roots were removed from pots and thoroughly rinsed in tap water. Disease severity (DS) was rated based on the percentage of affected stems and root system on a 1–5 scale, where 1 = 1 to 20%, 2 = 21 to 40%, 3 = 41 to 60%, 4 = 61 to 80% and 5 = 81 to 100% of stems showing chlorosis, necrosis or wilt (wilted stems were counted as DS 5) and of roots showing lesions [30]. The means and standard error of DS of stems and roots were calculated over the 9 replicates in total, i.e., from the three performed experiments each including three replications for each isolate and cultivar. After disease assessment, the plants were cut directly above the hypocotyl for fresh and dry weight estimation. Stems and roots were dried at 80 °C for 48 h. The means and standard error of fresh and dry weight were calculated over the nine replicates of each treatment and cultivar. 

In addition to the molecular detection of *Fusarium* spp. in plant tissue, one plant out of each replicate was used for the re-isolation of *Fusarium* spp. isolates on plates to fulfil Koch’s postulate. Root pieces (2–3 cm) were washed twice with tap water and dipped for 2 min in ethanol (2 mL, 70%) and thereafter dipped for 8 min in NaOCl (1 mL, 5% + Tween20) and finally rinsed with sterile water. The treated root pieces were then placed on PDA amended with streptomycin sulphate (50 mg L^−1^), penicillin-G-potassium (100 mg L^−1^) and tetracycline hydrochloride (10 mg L^−1^) and incubated at 25 °C for 2 days in the dark. *Fusarium* was determined by a visual and microscopic analysis of conidia [65].

### 4.6. Molecular Detection of Fusarium Spp. Isolates in Plant Tissue

A 1-cm piece of each stem was cut directly at the stem basis of each plant for pathogen DNA detection and gene expression analysis. Samples per plant were immediately frozen in liquid nitrogen and stored at −80 °C. Samples were taken from five plants per replicate of each treatment. Plant material was ground in liquid nitrogen in an orbital ball mill for 2 min at a frequency of 30 Hz s^−1^ (MM400 Retsch GmbH, Haan, Germany) with three balls of stainless steel with a 5-mm diameter.

Total DNA was extracted from 50 mg ground tissue using the DNeasy Plant Mini Kit (Qiagen GmbH, Hilden, Germany). DNA was quantified spectrophotometrically at 260 nm (Infinite M200PRO, Tecan Trading AG, Männedorf, Switzerland) using the NanoQuant plate, and quality was checked using absorption at 260 and 280 nm with a ratio >1.8 as acceptable. For the fungal DNA standard curves, mycelium was harvested from single-spore isolates grown for 5 days in liquid culture in PD medium in the dark at 25 °C. DNA from selected isolates per *Fusarium* spp. were extracted from 50 mg mycelium using the same methods. The standard curves were prepared using serial diluted samples of 2.5–0.001 ng of fungal DNA with and without 20 ng of pure *Asparagus officinalis* genomic DNA, thus resembling the plant background. Standard curves of y = 26.66 − 1.773x for *F. oxysporum* f.sp. *asparagi*, y = 26.16 − 1.758x for *F. proliferatum* and y = 26.34 − 1.770x for *F. redolens* were calculated in the presence of 20 ng of asparagus DNA. For qPCR detection of DNA, the primer concentrations of 200 nM and 3 µl of extracted DNA solution were used for each reaction in 10 µl with the Sensi Fast SYBR NO ROX Kit (Bioline GmbH, Luckenwalde, Germany). The qPCR cycling was carried out using cycle parameters of 95 °C for 5 min to activate the polymerase, followed by 40 amplification cycles of 95 °C for 15 s and 60 °C for 1 min with signal thresholds set automatically. Primers were based on sequence variations previously analysed by [7,26,27]. *F. oxysporum f.sp. asparagi*, Fo-RT-CL1f1: TTCATTTCTGCTGCTGAGCTTC, FoCL1r: GTCAGTAACTGGACGTTGGTACT; *F. proliferatum*, FpCL1f: CATGCATCAGACCACTCAAATC, Fp-RT-CL1r1: CGTTTAGCTCATGTTTTCGCTTC; *F. redolens*, FrEF1f: ATTTTCCCTTCGACTCGCCG, Fr-RT-EF1r2: GATCCGCGCTCATTGTGATTG.

### 4.7. Gene Expression Analysis in Stem Tissue Using RT-qPCR

RNA was extracted from 100 mg of ground tissue using the Qiagen RNeasy Plant Mini Kit including on-column DNaseI digestion (Qiagen, Valencia, CA, USA). RNA was quantified spectrophotometrically at 260 nm (Nanodrop ND1000, ThermoFisher Scientific, Schwerte, Germany) and quality controlled using a 2100 Bioanalyzer and RNA 6000 Nano kit (Agilent Technologies, Santa Clara, CA, USA) with a determining RNA integrity number (RIN). Single-stranded cDNA synthesis was carried out with 1 µg of total RNA using an iScript cDNA Synthesis Kit (Bio-Rad Laboratories GmbH, Feldkirchen, Germany) in a 25-µl reaction following manufacturer´s instructions, and subsequently diluted 10-fold. RT-qPCR was performed using 96-well reaction plates on a Thermal Cycler CFX96 C1000 Touch (Bio-Rad Laboratories GmbH) with the following thermal profile: 95 °C for 5 min, 40 cycles of 95 °C for 15 s and 60 °C for 1 min, followed by dsDNA melting curve analysis to ensure amplicon specificity. Each reaction was done in a 10-µL volume containing 200 nM of each primer, 3 µL of cDNA (1:10) and 5 µL of Sensi Fast SYBR NO ROX Kit (Bioline GmbH, Luckenwalde, Germany). The data were collected and compiled using CFX Manager Software 3.0 (Bio-Rad Laboratories GmbH). Satisfactory PCR amplification efficiency for each primer pair close to 2 was additionally verified using LinReg software [66]. Relative transcript levels were normalised on the basis of the expression of the invariant control elongation factor 1-alpha (XM_020411981), calculating ΔCq as the difference between control and target products (ΔCq = Cq*_gene_* − Cq*_EF1_*). Reference gene stability was calculated with the CFX Manager software 3.0 (Bio-Rad Laboratories GmbH) and also checked with qbase (Biogazelle, Gent, Belgium). At least three biological replicates of each cultivar inoculated with the respective *Fusarium* isolate were measured in duplicates, and non-template controls were included. For comprehensive presentation, the values of the respective species were presented together. Differences in relative expression levels between the treated samples were calculated as −ΔΔCq = ΔCq (*Fusarium* spp. inoculated sample) −ΔCq (mock inoculated sample). The oligonucleotide primer sets used for RT-qPCR are as follows: elongation factor 1-alpha (XM_020411981, *EF1*), Ao-RT-EF1alfa2f: TTGATAGGCGATCGGGTAAG, Ao-RT-EF1alfa2r: CTCATGTCCCTCACAGCAAA; pathogenesis-related protein 1 (XP_020276576, *PR1-2*), Ao-RT-PR1f2: TGTTCGAATCTGCCACTACT, Ao-RT-PR1r: TGCCTTCATGTGGTTGGTTA; pathogenesis-related protein 1-like (XM_020409857, *PR1-4*), Ao-RT-PR1f4: AGGCTTTTGTGATGGATTGG, Ao-RT-PR1r4: CTAGGCGCTCCTTGACGTAG; cationic peroxidase 1-like (XM_020420634, *POX*), Ao-RT-POX2f1: GCTTCAGCCCAGTTATCGTC, Ao-RT-POX2r1: CATTGACGAAGCAATCATGG; phenylalanine ammonia-lyase (XM_020404206, *PAL*), Ao-RT-PALf: GTAAACGACAACCCGCTCAT, Ao-RT-PALr: AGCTCCGATACCTGAGCAAA.

### 4.8. Analysis of the Root Morphology

The root systems of two randomly chosen plants of each replicate and cultivar of two experiments were used for imaging analysis. The data of the two experiments were pooled. To avoid overlapping, the washed roots were carefully separated on top of a DIN A4 sized bright frame and were imaged with a stage mounted digital camera (Canon EOS60 + EF 17–40 mm). Approximately 5 RAW images were obtained from each root tray with the following settings: aperture 10, ISO setting: 100 and focal length: 38 mm. The effective image resolution was about 450 DPI. A processing pipeline was realised by a MATLAB script (MATLAB. (2017) *version 9.3* (*R2017b*). The MathWorks Inc., Natick, MA, USA) that provided cleaned and margin-free binary root images to IJ-Rhizo (Rasband, W.S., ImageJ, U.S. National Institutes of Health, Bethesda, MD, USA, https://imagej.nih.gov/ij/, 1997–2018) [67] and performs additional pre- and post-processing steps. IJ-Rhizo was used to measure the root length of different diameter classes. 

This automated processing pipeline consisted of the following steps: keyboard macrobased RAW-file conversion (Digital Photo Professional, 2011), median filtering over a tif-converted raw image file stack (n~5 images), margin detection and removal via hough transform, root segmentation and dirt detection and removal. Root segmentation and dirt detection was based on fitted Gaussian mixture models and corresponding posterior probabilities. Obtained root diameter versus length histograms from IJ-Rhizo were cumulated over all subsamples per plant and then discriminated into fibrous and (thick) storage roots using the Otsu segmentation method [68]. Using this diameter threshold and adopting a cylinder geometry for roots, length, volumes and diameters were computed for both root categories from diameter versus length frequency table. Estimated root volumes showed a good correspondence (r^2^ = 0.74; *p* < 0.05, n = 30) to root fresh weight in a test set. 

### 4.9. Statistical Analyses

The data of DS were analysed by rank-based ANOVA according to the ANOVA-type statistics [69] using SAS PROC RANK and PROC MIXED with the ANOVAF option (SAS Institute 2016, Cary, NC, USA). Comparisons of means of the replicates (n = 9) were performed using χ^2^ test (*p* < 0.0001). Based on the multiple test procedure of comparisons of means *p* values were adjusted according to Holm [70]. Least-squares means were normalised to interval [0, 1] and used for correlation analyses with means of root traits (*p* < 0.05). The data of fresh and dry weight of roots and stems (n = 9), total root length, root volume, root surface area, and root radius (n = 6) were analysed by two-way ANOVA according to a randomized complete block factorial design. Comparisons of means of the replicates were performed using Tukey’s test (*p* < 0.05). 

For the statistical testing of the fungal utilisation of plant-derived polysaccharides, one-way ANOVA was applied. The identification of differences among the isolates within the substrate groups was performed with Tukey test (*p* < 0.05). Analyses of variance (ANOVA) were conducted using STATISTICA 13.5 software (StatSoft Inc., Tulsa, OK, USA). 

## Figures and Tables

**Figure 1 pathogens-09-00509-f001:**
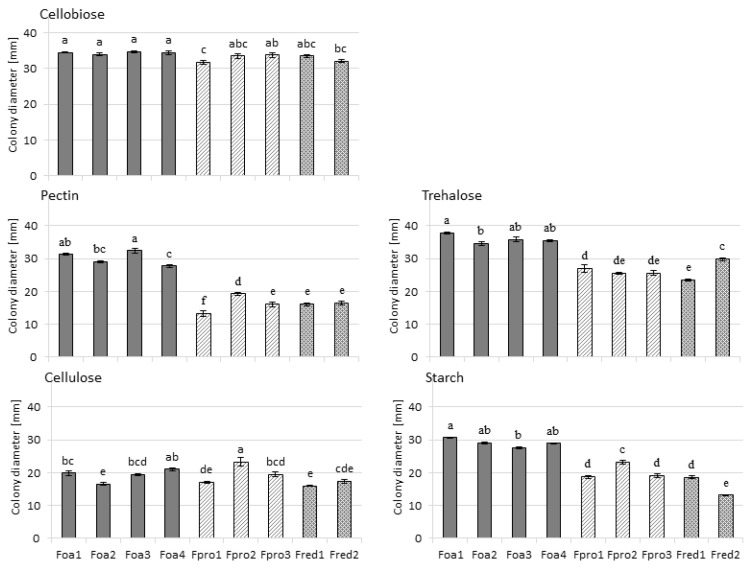
Colony diameter (mm) of *Fusarium* spp. grown on agar media containing plant-derived polysaccharides as a carbon source. Used *Fusarium spp.* are *F. oxysporum* f.sp. *asparagi* (Foa1, Foa2, Foa3, Foa4), *F. proliferatum* (Fpro1, Fpro2, Fpro3), and *F. redolens* (Fred1, Fred2); cell wall components (cellobiose, pectin, cellulose) are shown on the left, storage carbohydrates (starch, trehalose) are shown on the right. Mycelial diameters were measured at four days (cellobiose, trehalose, starch) and seven days (cellulose, pectin) after plate inoculation. The values are the means ± standard errors (n = 8). Significant differences are indicated by different letters (*p* < 0.05; Tukey test).

**Figure 2 pathogens-09-00509-f002:**
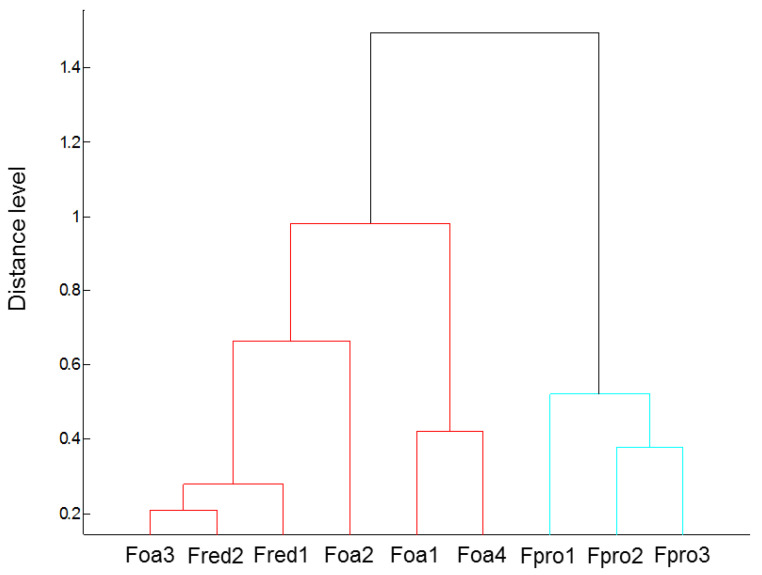
Cluster analysis of matrix-assisted laser desorption ionisation time-of-flight mass spectrometry (MALDI-TOF MS) spectra of *F. oxysporum* f.sp. *asparagi* (Foa1, Foa2, Foa3, Foa4), *F. proliferatum* (Fpro1, Fpro2, Fpro3), and *F. redolens* (Fred1, Fred2). Distance is displayed in relative units.

**Figure 3 pathogens-09-00509-f003:**
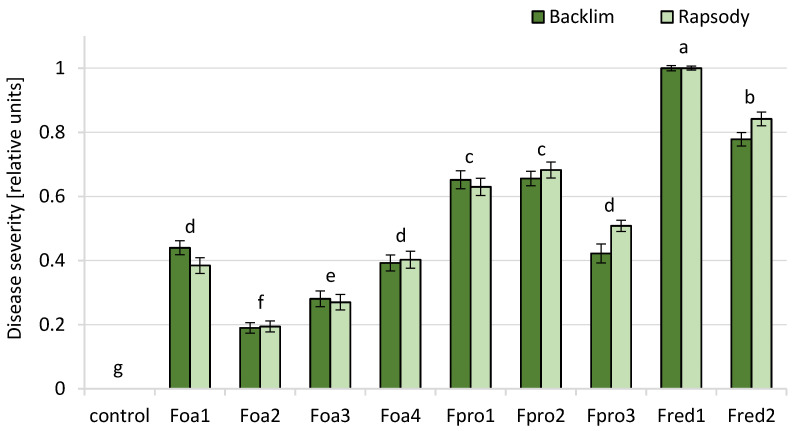
Disease severity (DS) of roots of asparagus ‘Backlim’ and ‘Rapsody’. Plants were inoculated at BBCH 12-13 with isolates of either *Fusarium oxysporum* f.sp. *asparagi* (Foa1, Foa2, Foa3, Foa4), *F. proliferatum* (Fpro1, Fpro2, Fpro3), *F. redolens* (Fred1, Fred2) with 6–8 × 10^6^ conidia mL^−1^, or with water (control) and sampled 8 weeks after inoculation (n = 9). The values are the means of ‘Backlim’ and ‘Rapsody’ ± standard errors of normalised ranks. The effect of isolates on plants was statistically analysed over both cultivars likewise; different letters show significance among the *Fusarium* spp. isolates and control (*p* < 0.0001; χ^2^ test).

**Figure 4 pathogens-09-00509-f004:**
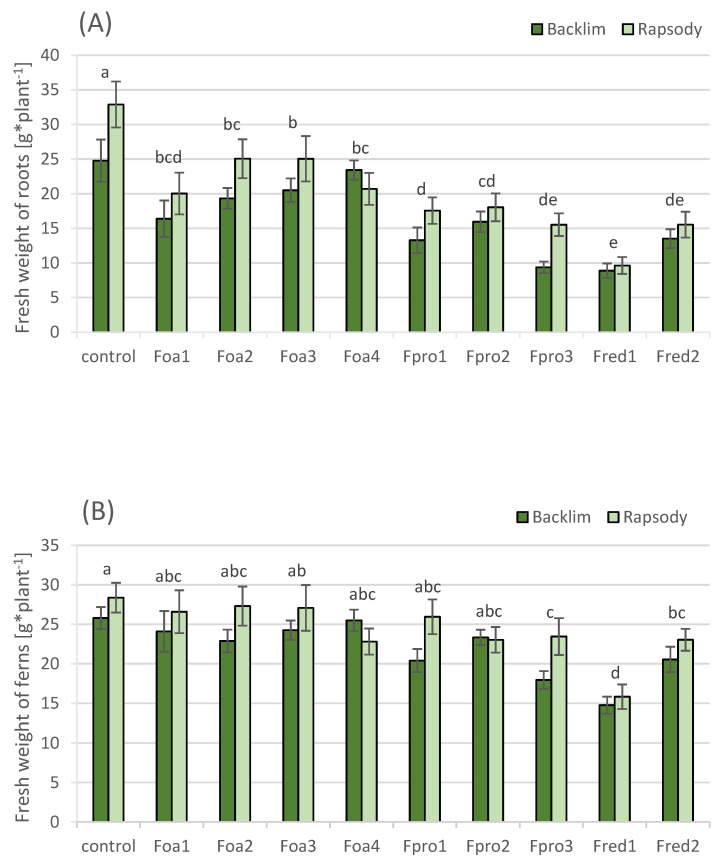
Fresh weights (g) of roots (**A**) and ferns (**B**) of asparagus ‘Backlim’ and ‘Rapsody’. Plants were inoculated at BBCH 12-13 with isolates of either *Fusarium oxysporum* f.sp. *asparagi* (Foa1, Foa2, Foa3, Foa4), *F. proliferatum* (Fpro1, Fpro2, Fpro3), *F. redolens* (Fred1, Fred2) with 6–8 × 10^6^ conidia mL^−1^, or with water (control) and sampled 8 weeks after inoculation (n = 9). The values are the means of ‘Backlim’ and ‘Rapsody’ ± standard errors. The effect of isolates on the fresh weight of plants was statistically analysed over both cultivars likewise; different letters show significance among the *Fusarium* spp. isolates and control (Tukey test; *p* < 0.05).

**Figure 5 pathogens-09-00509-f005:**
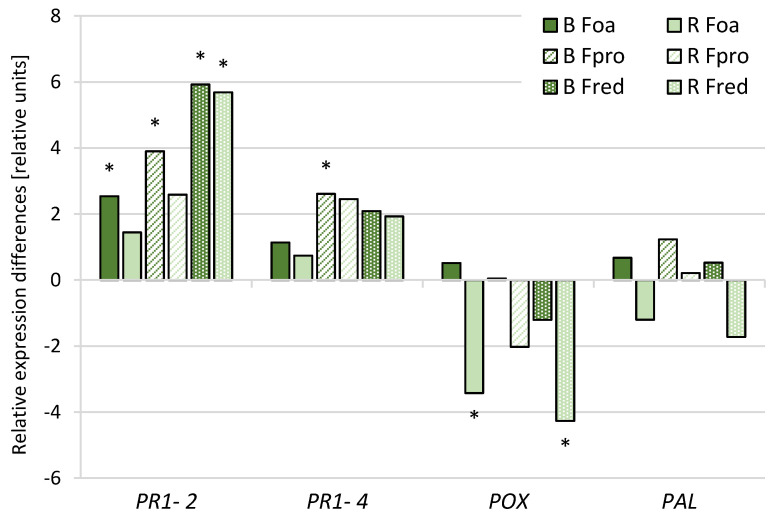
Gene expression analysis in stem bases of asparagus ‘Backlim’ (B) and ‘Rapsody’ (R). The plants were inoculated with isolates of *Fusarium oxysporum* f.sp. *asparagi* (Foa), *F. proliferatum* (Fpro), *F. redolens* (Fred), and water (control) (n ≥ 6). The values are the differences in relative expression levels between the means of treated samples, calculated as −ΔΔCq = ΔCq (*Fusarium* spp. inoculated sample) −ΔCq (respective control sample). * stands for significance between inoculated samples and control samples of the respective cultivar (*p* ≤ 0.05; *t*-test). *PR1-2*, pathogenesis-related protein 1 (XP_020276576,); *PR1-4*, pathogenesis-related protein 1-like (XM_020409857,); *POX*, cationic peroxidase 1-like (XM_020420634); *PAL*, phenylalanine ammonia-lyase (XM_020404206).

**Figure 6 pathogens-09-00509-f006:**
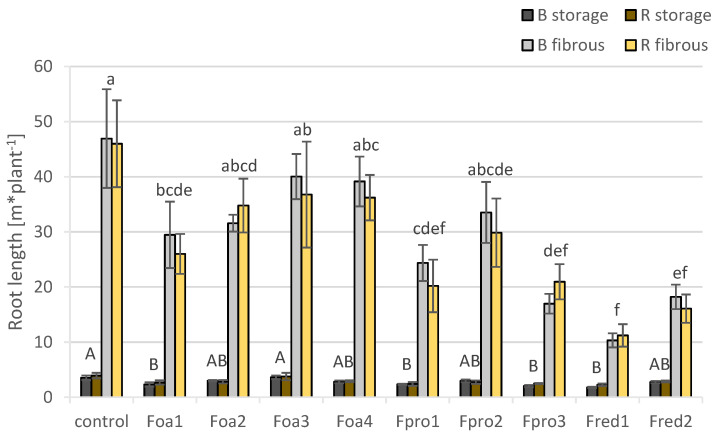
Total length of storage and fibrous roots (m) of asparagus ‘Backlim’ (B) and ‘Rapsody’ (R). Storage and fibrous roots were measured separately and summed up to total length (n = 6). Plants were inoculated with isolates of either *Fusarium oxysporum* f.sp. *asparagi* (Foa1, Foa2, Foa3, Foa4), *F. proliferatum* (Fpro1, Fpro2, Fpro3), *F. redolens* (Fred1, Fred2), or with water (control). The values are the means of ‘Backlim’ and ‘Rapsody’ ± standard errors. The effect of isolates on the total length of storage and fibrous roots was statistically analysed over both cultivars likewise; different letters show significance among the *Fusarium* spp. isolates and control (Tukey test; *p* < 0.05).

**Figure 7 pathogens-09-00509-f007:**
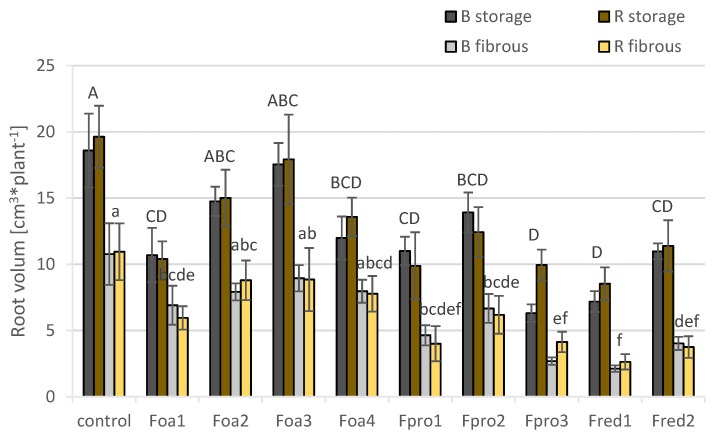
Root volume (cm^3^) of asparagus ‘Backlim’ (B) and ‘Rapsody’ (R). Plants were inoculated with isolates of either *Fusarium oxysporum* f.sp. *asparagi* (Foa1, Foa2, Foa3, Foa4), *F. proliferatum* (Fpro1, Fpro2, Fpro3), *F. redolens* (Fred1, Fred2), or with water (control) (n = 6). The values are the means of ‘Backlim’ and ‘Rapsody’ ± standard errors. The effect of isolates on root volume of storage and fibrous roots was statistically analysed over both cultivars likewise; different letters show significance among the *Fusarium* spp. isolates and control (*p* < 0.05; Tukey test).

**Figure 8 pathogens-09-00509-f008:**
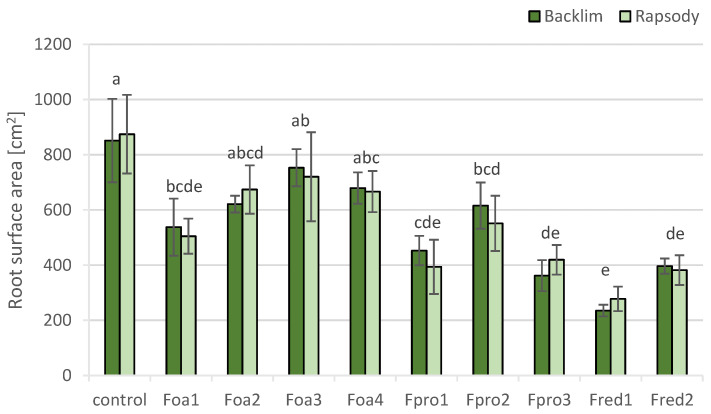
Root surface area (cm^2^) of asparagus ‘Backlim’ and ‘Rapsody’. Plants were inoculated with isolates of either *Fusarium oxysporum* f.sp. *asparagi* (Foa1, Foa2, Foa3, Foa4), *F. proliferatum* (Fpro1, Fpro2, Fpro3), *F. redolens* (Fred1, Fred2), or with water (control) (n = 6). The values are the means of ‘Backlim’ and ‘Rapsody’ ± standard errors. The effect of isolates on root surface area was statistically analysed over both cultivars likewise; different letters show significance among the *Fusarium* spp. isolates and control (*p* < 0.05; Tukey test).

**Table 1 pathogens-09-00509-t001:** Overview of *Fusarium. oxysporum* f.sp. *asparagi*, *F. proliferatum* and *F. redolens* isolates from asparagus from different German regions used for pathogenicity test, and molecular and physiological assays.

Isolate	Origin Code	Species	Origin	Isolation	Source *
Foa1	Ob2-66	*F. oxysporum*	Brandenburg	2002	HUB
Foa2	10K3.3	*F. oxysporum*	Niedersachsen	2001	HUB
Foa3	45K/2.1	*F. oxysporum*	Niedersachsen	2001	HUB
Foa4	III/3.1	*F. oxysporum*	Sachsen-Anhalt	2011	JKI
Fpro1	193-S	*F. proliferatum*	Brandenburg	2002	HUB
Fpro2	219-S	*F. proliferatum*	Rheinland-Pfalz	2000	HUB
Fpro3	88-17/1	*F. proliferatum*	Niedersachsen	2015	UAS
Fred1	89-17/2	*F. redolens*	Niedersachsen	2015	UAS
Fred2	90-7/1	*F. redolens*	Niedersachsen	2015	UAS

* HUB, Humboldt University of Berlin; JKI, Julius Kühn Institute Quedlinburg; UAS, University of Applied Sciences Osnabrück.

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
