# Peer review of "Species-Specific Impact of Fusarium Infection on the Root and Shoot Characteristics of Asparagus"

_pathogens, 2020, doi:10.3390/pathogens9060509_

Round 1

Reviewer 1 Report

I think this paper gives new information on Fusarium spp. isolates responsible for crown and root rot disease of  asparagus and underlines different agressiveness among Fusarium species.

I think that the research was well described and the paper is well written.

In particular, I suggest to the authors the following corrections:

- Page 1, line 13: please rewrite using cultivar Backlim and cultivar Rapsody.

- Page 1, line 41: what do you mean with “the ubiquitary model of life of Fusarium in the environment”?? Please modify this sentence trying to better explain what you are referring to.

 -Page 5, line 153: I think there is a mistake. You wrote: Root fresh weight of with Fusarium. Please make a correction.

- Page 14, lines 477-479: please remove specific description of the 2 cultivars used (with robust root system and compact fern and with a very vigorous growth). You repeated these information many times in results and discussion sections. You don’t have to write this again also in the materials and methods section.

Page 14, line 507: you speak about 9 replicates but in previous lines (500-501), you wrote that each experiment included 3 replications for each isolate and cultivar and that one replicate comprises 6 plants. I am a little confused about the 9 replicates (totally 54 plants). Can you better explain these numbers?

Author Response

Dear reviewer,

thank you very much for the fast revision.

We have made the following changes according to your suggestion:

  1. line 13, we changed the writing to "...analysed on the cultivar ‘Backlim’, with robust root system and compact fern, and the cultivar ‘Rapsody’, with a very vigorous growth." 
  2. line 41, we changed the wording to "...the ubiquitous presence of Fusarium in the environment..."
  3. line 153, corrected, we deleted the "with"
  4. lines 477 - 479, we deleted this specifc description of the cultivars in the Material and Methods section
  5. line 507, thank you very much for this point. We now clearly describe the numbers of replicates and the three repetitions of the experiment. We explained in lines 499 - 501 "The experiment was performed three times. Each experiment included three replications for each isolate and cultivar. One replicate comprised six plants of the respective cultivar." We now rewrote in line 507 "...were calculated over the 9 replicates in total, i.e. from the three performed experiments, each including three replications for each isolate and cultivar." We deleted here the number of analysed plants, because this was misleading with respect to the calculation of means and standard error.

With best regards

Rita Zrenner

Reviewer 2 Report

Dear Authors,

Please see in thext 1-2 small comments on your text.

Author Response

Dear reviewer,

thank you very much for your very fast revision.

We have included your suggestions as follows:

  1. line 138, we changed writing to did not
  2.  line 503, we inserted Disease severity (DS) in this chapter

With best regards

Rita Zrenner

Reviewer 3 Report

The paper describes a study on the Fusarium infection of two asparagus cultivars to show inter-specific differences among F. oxysporum f.sp. asparagi, F. proliferatum and F. redolens. The manuscript is reasonably well-written and it brings new data on Fusarium-asparagus interaction that might be of interest to broad international readership. Still, it requires a careful revision before it can be processed further. Some of my specific questions and suggestions have been outlined below.

The abstract is dull and does not give any attractive information. I suggest to re-write this section to emphasize the actual results obtained. Please, avoid general statements like “obvious differences were observed”. Instead, state clearly which isolate was the most aggressive compared to others.

Introduction section is concise and informative. I have only minor comments here: the name F. moniliforme (line 65) is no longer used. It was replaced by F. verticillioides. And, secondly, the aims of the study do not mention the shoot characteristics covered by the title of the manuscript.

Results are extensively and clearly presented. The description is also adequate. The same applies to the supplementary figures. I have no major questions regarding this section.

Discussion: a reference is needed for the statement given in lines 285-286. The paragraph given in lines 287-297 is weak. There is no clear description on what basis the identification is being done using MALDI-TOF MS technique and the other methods mentioned are chaotic and not accurate. I suggest to re-write this part of the Discussion. There are numerous examples of different approaches to identify and characterize Fusarium pathogens…

It is not that obvious that utilizing cell wall components as carbon sources by the pathogen strains directly reflects their aggressiveness. The Authors should also check the activity of the enzymes responsible for degradation of the cell wall components.

The paragraph given in lines 374-383 is actually a repetition of the results summary. Please, re-phrase it or consider deleting this part.

There is one thing in the Discussion that is missing: mycotoxin consideration. Fusarium species are known for their mycotoxigenic abilities. There is no way that the mycotoxins produced do not play a role in the plant-pathogen interaction. The Authors have completely neglected this issue and this is a major flaw. At least the differences in metabolic profiles of the three species tested should be discussed, even, if the Authors did not measure the levels of BEA, ENNs, FBs and MON (to mention just the basic ones) produced by the species used.

Materials and Methods: the species identity of the strains used in the experiments should be verified either by morphological or molecular methods, or at least a reference should be cited to this part. The method mentioned in the re-isolation description (lines 512-518) is also not sufficient.

Again, it should be clearly stated that the MALDI-TOF MS species discrimination was done based on the metabolite (?) or protein (?) fingerprinting.
